# The Gender Gap in Job Status and Career Development of Chinese Publishing Practitioners

Yawen Li [1] and Yushan Zhao [1,2,*]

1   School of Journalism and Communication, Beijing Normal University, Beijing 100089, China
2   Institute of Publishing Science, Beijing Normal University, Beijing 100875, China
*   Correspondence: 03089@bnu.edu.cn

**Abstract:** There is a significant difference between the number of male and female Chinese publishing practitioners. To investigate the gender gap among Chinese publishing practitioners, we surveyed 3372 valid questionnaires from 30 April 2020 to 31 December 2020. This research mainly adopts the Chi-square and T-test to analyze the gender gap in publishing practitioners' career choices, career plans, career developments, etc. The results show that although females occupy nearly 70% of the samples in the data, males perform more competitively in multiple indicators such as salary and career development. There is also a significant gender gap in terms of career plans and career perception. However, our research shows that the gender gap is not obvious in terms of workload and willingness to change jobs. This study provides a factual basis and data support for the current gender situation, and discusses the reasons for the gender gap in the Chinese publishing industry, to provide references for the Chinese publishing industry to build a healthy gender structure.

**Keywords:** gender gap; publishing practitioners; job status

## 1. Introduction

### 1.1. Motivation and Objectives

With the progress of society, more and more females participate in economic activities and work, and their social status has gradually improved. Gender equality has become a hot topic among people, and many intellectuals are advocates of gender equality. However, gender-based discrimination and injustice still exist, and a study has shown that the COVID-19 pandemic has amplified gender gaps. Between March 2020 and September 2021, females were more likely to report employment loss than males, as well as report forgoing work to care for others. Females and girls were more likely than males and boys to report dropping out of school for reasons other than school closures [1]. With a recent USA, Today headline proclaiming "Sexism in the Workplace Is Worse Than You Thought" [2], it is clear that gender bias remains a challenge for females in the workplace, especially as they try to move up the career ladder. Professional women face an invisible obstacle that hinders their leadership ambitions. The accepted term commonly used to describe this plight of females in the workforce is the glass ceiling, which refers to the invisible barrier that females face as they advance through the ranks [3]. Females cannot be promoted to the top of the enterprise or organization, not because of their lack of ability or experience or because they do not want to obtain these positions, but because gender stereotypes hold that females are not suitable for leadership roles because they have more collective characteristics than active ones [4]. The segmentation theory in sociology emphasizes that there are various segmentation factors in the labor market and in society. This structural factor will affect the acquisition of personal status. At the same time, family background and other pre-endowing factors become the factors that determine which kind of labor market an individual enters [5]. Theodore Schultz explained gender differentiation in careers using the theory of human capital investment, and that the reason why human

capital investment is inclined to the male is that it is more valuable to invest in males than in females [6]. Given the social structure and the culture of people, on average, female entrepreneurs have more family-related obligations than males (raising and looking after children, devoting time to household chores as a married woman, etc.). These facts can create time and mobility constraints when it comes to starting or developing a business [7]. These theories are combined to form part of the basis of the phenomenon of the glass ceiling. Although researchers have shown that creating a gender-balanced business environment is realized by enforcing equality in capacity, propriety rights, and employment, sustainable businesses derive great benefits from such an institutional context [8]. Many governments are trying to improve the relevant laws and policies, but various factors support and, in some cases, strengthen the gender gap. The gender gap in career development, including, but not limited to, the glass ceiling, is still prominent.

In 2020, females accounted for 43.5% of China's labor force, and males accounted for 56.5%. The Global Gender Gap Report measures the gender gap from four dimensions: Economic Participation and Opportunity, Educational Attainment, Health and Survival, and Political Empowerment. In 2022, the global gender gap was closed by 68.1%, but no country has yet achieved full gender parity. At the current rate of progress, it will take 132 years to reach full parity. In 2022, China closed 68.2% of its gender gap, ranking 102nd out of 146 economies in The Global Gender Gap Index 2022 rankings [9]. According to the data of the fourth survey on the social status of Chinese females, with the further optimization of the social environment for promoting gender equality and females' comprehensive development in China, females play an important role in economic and social development, and the power of "half the sky" is further highlighted [10].

In the context of the increasing number of working females, this paper focuses on the current status of the gender gap in the female-dominated publishing industry in China. In the publishing industry of China, where the number of females is significantly higher than that of males, does the gender gap still affect the job status of publishing practitioners? Are the jobs equal for males and females? Who controls the publishing industry? Do females get paid the same as males? Do females and males have equal opportunities for promotions? Do they have the same expectations for career development? Are females in the publishing industry still unable to break the glass ceiling?

This paper will answer the above questions through questionnaire data. At the same time, we will explore the reasons for the gender gap in the job status of Chinese publishing practitioners, and make reasonable suggestions for the relevant management departments and publishing institutions in China.

*1.2. Related Work*

The gender gap at work is worldwide; among the 190 economic problems analyzed by the World Bank, females only enjoy 75% of the legal rights and interests of males [11]. European data show that, despite a significant increase in female participation in education, the labor market and political life over the past few decades, females are still paid less than males and are underrepresented at supervisory, managerial and executive levels [12]. Empirical evidence shows that there is a significant gender gap in terms of the impact on overall productivity and in academic careers in the fields of science, technology, engineering and mathematics [13]. Gender gaps are not only between industries, but gender gaps within the same industry are also obvious. The study found that females are only half as likely to be promoted to entrepreneurship partner roles as males [14]. Scholars have studied the gender differences in entrepreneurship and intrapreneurship, and found that females are less likely to choose entrepreneurship, presumably because of their aversion to risk, the existence of credit constraints or discrimination [15]. An analysis of biochemist survey data gathered in 1981 shows that females have a lower probability of being promoted to professors or associate professors [16]. Carr et al. [17] found that, in a survey of nearly 2000 faculty members from 24 academic medical centres that made comparisons with male faculty members or female faculty members without children, female faculty members

with children have less secretarial support, less institutional research funding and lower job satisfaction.

As for the publishing industry, a survey on female jobs, wages and gender discrimination in the US publishing industry shows that the wages of females in publishing houses in the US are lagging behind males'. Although females have performed well in their previous jobs, they more easily miss promotions. The real authority is in the hands of males [18]. Females occupy the vast majority of positions in the UK publishing industry, but they lack an advantage in terms of pay [19]. Among the publishing practitioners in China, the number of female employees is significantly higher than that of males, which seems to be a female-friendly industry. However, existing studies find that the advantage of females in numbers does not represent an advantage in the workplace. For example, it is demonstrated that the leaders of the vast majority of publishing institutions in China are male [20]. It is believed that gender discrimination still exists in the process of job hunting and promotion for females in the publishing industry, and it is difficult for females in the workplace to obtain the same opportunities for promotion and salary increment as males [21].

There are not many studies on the gender gap in the publishing industry in China, and most of the existing studies focus on a descriptive analysis of job status and feelings. There is little exploratory research on the gender gap in the publishing industry through data analysis and comparison. Therefore, this paper searches for existent evidence of the gender gap through quantitative data and strives to provide a factual basis and data support for the healthy development of gender construction in China's publishing industry.

## 2. Materials and Methods

### 2.1. Data Collection

The survey was conducted by the Academy of Publishing Sciences of Beijing Normal University, and was released through the WeChat public account of "Mu duo shu sheng". "Mu duo shu sheng" is affiliated with the Academy of Publishing Sciences of Beijing Normal University; it is dedicated to providing readers with professional knowledge and industry news in the publishing industry, as well as building a platform and academic support for publishing practitioners to learn and communicate. It is an authoritative official public account that is recognized and well-known by publishing practitioners. From 30 April 2020 to 31 December 2020, a total number of 3732 valid questionnaires were received.

The sample covers state-owned publishing houses, newspaper publishers, private companies, and other publishing units. The employees of state-owned publishing houses still account for the majority, accounting for 56.4%. The survey samples cover 34 provinces, autonomous regions and municipalities in China. The number of samples from Beijing was the largest, up to 1306, accounting for 34.99% of the total number of samples. This is closely related to the fact that Beijing, as the national cultural center, has a large number of publishing institutions and publishing practitioners. In terms of geographical distribution, North China has the largest number of samples, and the number of samples from other regions is from high to low in the order of East China, Central China, South China, Northwest China, Northeast China, etc. (Table 1).

The questionnaire was divided into six parts, including basic information, workload and working time, salary and welfare, career growth, career choice, and work pressure, with a total of 54 questions. Relevant quantitative indicators such as stress and satisfaction were set with a 10-level scale. Numbers ranging from 1 to 11 correspond to answers ranging from "strong" to "weak": 1 is "very weak", 6 is "average", and 11 is "very strong". Other scales or codes are marked in the tables involved and can be viewed on the data open website. In the questionnaire, we selected questions related to the gender gap for analysis. Then, we reorganized these into five categories: basic information about publishing practitioners, goals and workload, pay and benefits, career choices and job perception.

**Table 1.** Statistical table of sample regions and units.

| Regions | no. | % |
|---|---|---|
| North China | 1621 | 43.44% |
| East China | 764 | 20.47% |
| Central China | 378 | 10.13% |
| South China | 264 | 7.07% |
| Northwest China | 246 | 6.59% |
| Northeast China | 246 | 6.59% |
| Southwest China | 213 | 5.71% |
| Total | 3732 | 100% |
| **Units** | **no.** | **%** |
| Newspaper publishers-Science and Technology | 217 | 5.81% |
| Newspaper publishers-Social Sciences and Education | 223 | 5.98% |
| Newspaper publishers-General Public | 136 | 3.64% |
| University Press | 646 | 17.31% |
| Provincial (city) local publishers | 821 | 22.00% |
| Central part-owned publishers | 638 | 17.10% |
| Planning and publishing companies (private) | 492 | 13.18% |
| Distribution and promotion company (private) | 43 | 1.15% |
| Typesetting and proofreading companies (private) | 105 | 2.81% |
| New media technology company (private) | 50 | 1.34% |
| Other companies | 361 | 9.67% |
| Total | 3732 | 100% |

*2.2. Data Analysis*

This research mainly adopted the Chi-square test and the T-test for significance analysis. The statistical methods used to study the relationship between the variables were the Chi-square test, T-test and variance analysis. In this study, the independent variable was gender, which is a categorical variable that includes male and female. The dependent variables consisted of both categorical and quantitative variables. Therefore, the categorical dependent variables included the following: education, professional title, job series, etc.; we use the Chi-square test to explore the gender gap. The dependent variables, which are quantitative, include the following: business trip days, stress, satisfaction with salary and benefits, etc.; we use the T-test to explore the impact of the gender gap.

**3. Results**

According to the survey data, there were 2592 females (69.45%) and 1140 males (30.55%). The data show that females still occupy the majority, which is consistent with the actual situation of the publishing industry.

### 3.1. The Gender Gap between Publishing Practitioners in the Job Status

Table 2 shows that, in the sample, there is a significant gender gap in terms of several basic indicators such as age, educational background, years of publishing work, and job title ($p < 0.01$, which means significant).

**Table 2.** Chi-square test for gender gap w.r.t basic situation.

| | Male No. (%) | Female No. (%) | Total No. (%) | *p* |
|---|---|---|---|---|
| **Age** | | | | |
| ≤25 | 43 (3.77) | 177 (6.83) | 220 (5.89) | |
| 26~30 | 212 (18.60) | 608 (23.46) | 820 (21.97) | |
| 26~35 | 223 (19.56) | 652 (25.15) | 875 (23.45) | |
| 36~45 | 381 (33.42) | 862 (33.26) | 1243 (33.31) | 0.000 ** |
| 46~55 | 209 (18.33) | 257 (9.92) | 466 (12.49) | |
| 56~60 | 57 (5.00) | 30 (1.16) | 87 (2.33) | |
| ≥61 | 15 (1.32) | 6 (0.23) | 21 (0.56) | |
| Total | 1140 (100.00) | 2592 (100.00) | 3732 (100.00) | |
| **Educational background** | | | | |
| Junior high school | 3 (0.26) | 0 (0.00) | 3 (0.08) | |
| Higher school | 6 (0.53) | 1 (0.04) | 7 (0.19) | |
| College | 67 (5.88) | 103 (3.97) | 170 (4.56) | |
| Bachelor | 587 (51.49) | 1146 (44.21) | 1733 (46.44) | 0.000 ** |
| Master | 425 (37.28) | 1252 (48.30) | 1677 (44.94) | |
| Doctor | 52 (4.56) | 90 (3.47) | 142 (3.80) | |
| Total | 1140 (100.00) | 2592 (100.00) | 3732 (100.00) | |
| **Year of publishing work** | | | | |
| ≤5 | 331 (29.04) | 982 (37.89) | 1313 (35.18) | |
| 6~10 | 254 (22.28) | 684 (26.39) | 938 (25.13) | |
| 11~20 | 364 (31.93) | 730 (28.16) | 1094 (29.31) | 0.000 ** |
| 21~30 | 138 (12.11) | 174 (6.71) | 312 (8.36) | |
| ≥31 | 53 (4.65) | 22 (0.85) | 75 (2.01) | |
| Total | 1140 (100.00) | 2592 (100.00) | 3732 (100.00) | |
| **Job title** | | | | |
| None | 264 (23.16) | 679 (26.20) | 943 (25.27) | |
| Primary professional title | 95 (8.33) | 226 (8.72) | 321 (8.60) | |
| Intermediate professional title | 434 (38.07) | 1203 (46.41) | 1637 (43.86) | 0.000 ** |
| Deputy Senior professional title | 253 (22.19) | 408 (15.74) | 661 (17.71) | |
| Senior professional title | 94 (8.25) | 76 (2.93) | 170 (4.56) | |
| Total | 1140 (100.00) | 2592 (100.00) | 3732 (100.00) | |

** $p < 0.01$.

In terms of age, the publishing industry practitioners in the samples were mainly young people aged 26 to 45. Nearly 90% of females and 75% of males were under the age of 45. The age of 45 has become a watershed for the gender gap. Above the age of 45, the proportion of males in the samples exceeds females, i.e., the gender gap gradually becomes balanced as the working age increases. This phenomenon should be attributed to the different retirement ages of males and females.

As for the educational background, the minimum education level for females is high school, while it is junior high school for males. More than 50% of females hold a master's degree or above, while the figure for males is merely 42%.

As for the working experience, females who have worked for less than 10 years account for 64.28% while males account for 51.32%. The T-test results demonstrate that the average working years of males is higher than that of females. Therefore, female publishers are younger and have shorter career lifespans than males.

As a whole, the number of females in the sample is 2.33 (2592/1140) times that of males. Let us look at the proportion of females and males in different levels of professional titles. The number of females without a professional title is 2.58 (679/264) times that of males, the number of females with a primary professional title is 2.38 (679/264) times that of males, the number of females with an intermediate professional title is 2.77 (1203/434) times that of males, and the number of females with a deputy senior professional title is reduced to 1.61 (408/253) times that of males. In senior professional titles, the number of males is 1.28 (97/76) times higher than that of females. The data show that female publishing practitioners continue to be at a disadvantage when they are promoted: in middle and low-level positions, females do not have obvious disadvantages in their career development, but when they are promoted to the deputy senior level, females begin to have disadvantages in their professional titles; meanwhile, in the competition for senior positions, females' disadvantages are very obvious. It can be seen that female publishing practitioners also face a serious glass ceiling in their career development.

According to the sample conclusion, we can speculate that, compared with males, females in the publishing industry have a higher population, higher education, fewer working years, and lower job titles. Females make up the majority of publishing practitioners, however, males dominate in job promotions.

### 3.2. The Gender Gap in Goals and Workload

The data show that females in "content editing" positions account for 85.61% and males account for 72.81%. Females in "administrative management" positions account for only 10.53% and males account for 19.82%, which means that the gap is obvious. The proportion of females in "marketing and distribution" positions is lower than that of males. There is no significant gender gap in technical positions (less than two percentage points).

Considering future career plans, nearly 60% of females' ideal career goal is to become a technical expert, while the figure for males is only 50%. Less than 20% of females' ideal goal is to become a senior manager, and the remaining 20% of females are willing to adjust their career development goals. It can be seen that females are more willing to focus on technical positions, while males are more willing to be engaged in management positions (Table 3).

**Table 3.** Chi-square test for gender gap w.r.t career choices.

| | Male No. (%) | Female No. (%) | Total No. (%) | *p* |
|---|---|---|---|---|
| **Job positions** | | | | |
| Administration | 226 (19.82) | 273 (10.53) | 499 (13.37) | |
| Content editing | 830 (72.81) | 2219 (85.61) | 3049 (81.70) | |
| Marketing distribution | 61 (5.35) | 64 (2.47) | 125 (3.35) | 0.000 ** |
| Technical Services | 23 (2.02) | 36 (1.39) | 59 (1.58) | |
| Total | 1140 (100.00) | 2592 (100.00) | 3732 (100.00) | |
| **Career plans** | | | | |
| Junior technician to technical expert | 517 (45.35) | 1330 (51.31) | 1847 (49.49) | |
| Junior manager to technical expert | 102 (8.95) | 187 (7.21) | 289 (7.74) | |
| Junior manager to senior manager | 106 (9.30) | 125 (4.82) | 231 (6.19) | 0.000 ** |
| Junior technician to senior manager | 205 (17.98) | 367 (14.16) | 572 (15.33) | |
| Work hard, based on needs | 210 (18.42) | 583 (22.49) | 793 (21.25) | |
| Total | 1140 (100.00) | 2592 (100.00) | 3732 (100.00) | |

** $p < 0.01$.

Four indicators were selected to measure workload and working time in the survey: overtime hours in a week, business trip days in a year, writing workload completed in a year, and general workload. These four indicators are all quantitative variables, so we use the independent sample T-test to estimate the confidence of the gender gap. The T-test results (Table 4) showed that there were small but significant differences in overtime hours in a week and business trip days in a year ($p < 0.05$), i.e., females' results were slightly lower than males. However, there was no significant difference in terms of the amount of text processing (after removing invalid samples) and the overall perceptual workload ($p > 0.05$, not significant at the level of 95%).

**Table 4.** T-test for the gender gap in workload.

| Variable | Encoding | Gender (Mean ± SD) | | T | p |
| --- | --- | --- | --- | --- | --- |
| | | **Male (n = 1140)** | **Female (n = 2592)** | | |
| **Overtime hours per week** | 1–6: no overtime to over 30 h | 2.90 ± 1.35 | 2.73 ± 1.21 | 3.556 | 0.000 ** |
| **Days on the business trip per year** | 1–6: no business trip to business trip for more than 30 days | 3.10 ± 1.66 | 2.39 ± 1.42 | 12.559 | 0.000 ** |
| **Current workload** | 1–11: easy to do to, hard to do | 7.06 ± 2.41 | 7.19 ± 2.21 | −1.492 | 0.136 |
| **Variable** | **Encoding** | **Male (n = 797)** | **Female (n = 2134)** | **T** | **p** |
| **Text processing requirements** | 1–7: no such indicator to more than 10 million words | 2.59 ± 1.73 | 2.57 ± 1.77 | 0.264 | 0.792 |

** $p < 0.01$.

The above data show that there is a significant gender gap in workload and career plans in the sample data. Females focus more on content editing, and their ideal goal is to become technologists. The majority of males also work in content editing roles, but males are more willing to work in executive management, and their ideal goal is to become senior executives. In terms of manual labor indicators, such as working hours, overtime, and business trips, males show more advantages due to physiological differences.

*3.3. The Gender Gap in Pay and Benefits*

Table 5 shows that 28.28% of females and 42.89% of males have a monthly average after-tax income of more than RMB 8000, the latter of which is much higher than that of females. In total, 15.36% of females and 21.66% of males have a year-end bonus of more than RMB 50,000. Although the number of female samples is nearly 2.3 times larger than that of males, more males have a monthly salary of more than RMB 12,000 and a year-end bonus of more than RMB 200,000. Overall, there are fewer high-paid females in the publishing industry, and females are paid less than men.

**Table 5.** T-test for the gender gap in salaries and benefits.

| Variable | Encoding | Gender (Mean ± SD) | | T | p |
| --- | --- | --- | --- | --- | --- |
| | | **Male (n = 1140)** | **Female (n = 2592)** | | |
| Monthly salary | 1–8: less than RMB 3000 to more than RMB 30,000 | 3.44 ± 1.35 | 3.04 ± 1.09 | 9.612 | 0.000 ** |
| Expect monthly salary | 1: it is more reasonable at present, 2–8: RMB 3000 to RMB 30,000 RMB or more | 4.66 ± 1.67 | 4.18 ± 1.47 | 8.254 | 0.000 ** |
| Income among local workers | 1–3: low, medium, high | 1.65 ± 0.61 | 1.47 ± 0.54 | 8.948 | 0.000 ** |
| Year-end bonus after tax | 1–8: RMB 0 to 200,000 or more | 3.19 ± 1.69 | 2.92 ± 1.47 | 5.003 | 0.000 ** |
| Salary and benefits Satisfaction | 1–11: very dissatisfied to very satisfied | 6.07 ± 2.23 | 5.69 ± 2.12 | 5.006 | 0.000 ** |
| Salary package among local peers | 1–11: not at all competitive to very competitive | 4.83 ± 2.02 | 4.24 ± 1.91 | 8.543 | 0.000 ** |

** $p < 0.01$.

Compared to the average earnings of local employees, only 2.35% of females and 7.11% of males reported that they have higher actual earnings. In total, 24.74% of females and 36.32% of males believe that their salary is higher than that of their peers (measurement encoding ≥ 6). Compared with the expected monthly salary, only 5.35% of females and 6.06% of males believe that their actual monthly salary (including provident fund, excluding year-end bonus) is reasonable. Most practitioners expect a monthly salary between RMB 8000 and 15,000. However, the number of females (17.9%) who have a monthly salary of more than RMB 15,000 ("considered reasonable") is significantly lower than males (30.61%). It can be seen that females' income expectations are significantly lower than males. In addition, in terms of the practitioners' satisfaction with their average wages and benefits, 33.59% of females and 42.9% of males are relatively satisfied (scores ≥ 6). Females are also less satisfied than males. It can be seen that, although the salary expectations of female practitioners are lower than that of males, the publishing industry is still not recognized by more female practitioners in terms of salary satisfaction.

Since several indicators of wages and benefits are quantitative variables, we used the independent samples T-test to detect the gender gap of them. The *p*-values of each index are all less than 0.01, indicating that there is a significant gender gap in the five variables: monthly mean income, year-end after-tax bonus, salary and welfare satisfaction. The average value of these five variables for females is lower than for males. The average real income of females is lower than that of males, and fewer females are paid more than males. Meantime, the expected income value of females is also lower than that of males. Publishing practitioners feel underpaid compared to local employees. Compared with their local counterparts, most publishing practitioners consider compensation and benefits to be at a disadvantage and relatively non-competitive.

*3.4. The Gender Gap in Career Choices*

It can be seen from the survey data that "Like books, love publishing", "matching of major" and "the stability of state-owned enterprises" are the main reasons that most people enter the publishing industry. In total, 61.77% of females and 66.40% of males entered the publishing industry because they preferred the publishing industry; 25.73% of females and 22.28% of males entered because of their majors; 25.12% of females and 22.63% of males entered because they valued stability and life-long guaranteed jobs in state-owned enterprises; and 10.76% of females and 12.89% of males entered because they would have relatively more free time after work. It can be seen that females pay more attention to matching their major to their job and job stability, while males pay more attention to hobbies and free time after work.

From the data, 58.41% of females and 43.95% of males have only worked in one department; and 3.74% of females and 8.94% of males have worked in more than 3 departments. When facing their career plans, the proportion of males who identify with "cross series" is significantly higher than that of females (Table 2). It can be seen that the job flexibility of female employees is lower than that of males. In terms of career plans, females are more willing to insist on only one position, while males are more inclined to try more positions.

The responses of males and females to the integration of new technologies in publishing are also quite different. Among females, 57.3% of them choose to "rely on their current skills" and "be mentally prepared, but have no idea where to start". However, only 49.3% of males choose this option. This shows that when facing new technologies, males are more active in learning and exploring than females. On the whole, because of the possibility to work in more positions and because they are more willing to explore career plans, the diversity of career development in males is better than that of females.

There is a little gender gap in the willingness to change jobs. In total, 44.76% of females and 45.44% of males have plans to change jobs. Females and males will give priority to the following options: remain in the publishing industry, education industry or government-related departments. There is no significant difference in the reasons and motivations for job-hopping. The top five reasons for job-hopping are dissatisfaction with

salary and benefits, disapproval of the current corporate cultures, pessimism about the future of the publishing industry, leadership issues and personal reasons (family, health, etc.). The main reasons why females choose to stick to the publishing industry and do not want to change jobs are "high life pressure and high risk of job-hopping" (18.09%) and "personally like the fragrance of books, but do not like challenges" (17.28%). Male's choices are "full confidence in the cultural publishing industry" (22.11%) and "high life pressure and high risk of job-hopping" (18.07%). The third common reason is that "the current development is relatively satisfactory and that they are unwilling to take risks". In terms of the motivation for job-hopping, it can be seen that the priorities of males are the objective development environment, life pressure and career development. Female priorities are life pressures, personal feelings and career development, respectively.

Judging from the results of the above career choices, in terms of reasons for entering the industry, males pay more attention to hobbies than females, and females pay more attention to the matching of their major than males; females focus on job stability, and males focus more on their free time. After entering the workplace, females are more stable and more willing to specialize in one position, while males are more willing to develop in an all-around way. Although there is no basic difference in the willingness of males and females to change jobs, females pay more attention to personal feelings and males pay more attention to the overall situation of the industry.

### 3.5. The Gender Gap in Job Perceptions

There is a significant gender gap in their attitude towards future career prospects and the clarity of career plans ($p < 0.01$). The average score for their future career prospects was 6.23 for females and 6.57 for males. For clarity of career plans, the average score for females is 6.85 and 7.30 for males. As for these two indicators, the scores of females are slightly lower than for males, and it can be seen that the clarity of careers plans for females is slightly lower than for males (Table 6). The satisfaction with the job status of females is significantly lower than that of males. It can be speculated that in the publishing industry, compared with females, males consider themselves more professional and competitive. Males have a higher clarity of career plans and confidence in their career prospects than females.

**Table 6.** T-test for the gender gap in career emotions.

| Variable | Encoding | Gender (Mean ± SD) | | T | *p* |
|---|---|---|---|---|---|
| | | Male(n = 1140) | Female (n = 2592) | | |
| **Career prospects** | 1–11: very pessimistic to very optimistic | 6.57 ± 2.19 | 6.23 ± 1.88 | 4.591 | 0.000 ** |
| **Career plans** | 1–11: very unclear to very clear | 7.30 ± 2.11 | 6.85 ± 1.96 | 6.141 | 0.000 ** |
| **Work attitude** | 1–4: I like it very much to I do not like it | 2.16 ± 0.67 | 2.22 ± 0.65 | −2.467 | 0.014 * |
| **Work pressure** | 1–11: very small to very large | 7.62 ± 2.07 | 7.57 ± 1.93 | 0.719 | 0.472 |
| **Job satisfaction** | 1–11: very dissatisfied to very satisfied | 6.81 ± 2.00 | 6.48 ± 1.82 | 4.863 | 0.000 ** |

\* $p < 0.05$ ** $p < 0.01$.

In terms of work attitudes, the difference is clear, but not significant. In total, 68.44% of females and 71.75% of males "strongly prefer their job" and are "somewhat satisfied". This means that about 70% of publishing practitioners still prefer the publishing industry. Publishing practitioners have a high level of loyalty to the publishing industry.

As for work stress, there is no significant gender gap ($p > 0.05$), and the gender stress values for both males and females are around 7.6. In the professional experience of both males and females, it is honorable to be a recognized expert in the publishing industry. They are frustrated when facing their career prospects and are unable to improve their skills. There is no significant gender gap in the judgment criteria of career values.

Judging from the T-test results of the gender gap in terms of career sentiment among publishing practitioners, although males and females share the same standards and values in career honor and frustration, males are more confident and competitive than females in

career plans and career prospects. Although females are at a disadvantage compared to males in career development, males and females exhibit the same work attitudes and stress.

## 4. Conclusions

Through the analysis of the survey data, it can be seen that the gender gap affects the job status and career development of publishing practitioners in many aspects, including the following results:

In general, the number of male and female employees in the data is 3:7. The publishing industry remains a knowledge and information-intensive industry that is dominated by female employees. Compared with male employees, female employees were more highly educated, had a lower working age and had lower professional titles.

From the perspective of career development, in terms of the motivation to enter the publishing industry, females pay more attention to the matching of their major, while males pay more attention to their hobbies. Females focus more on the stability of work, while males focus on their free time after work. There was no difference in the intensity of the willingness to change jobs between males and females. In terms of whether to consider changing jobs, females focus on personal feelings, while males focus on the overall situation of the industry.

From the perspective of career development goals and plans, content editing and proofreading positions in the publishing industry are the core positions, and more than 70% of the samples are engaged in such jobs. Compared with males, females' positions are more stable, and they are more willing to stick to their familiar positions and look forward to becoming technical experts. Compared with females, males are more willing to take up management positions, have strong job mobility, and are more willing to explore and try diversified businesses. The ideal career plan is to become a senior manager.

From the perspective of remuneration and career satisfaction, although there is a small gender gap in terms of workload, the average actual income of females is lower than that of males, and the number of females with high salaries is also less than that of males. Males are more competitive than females in future job prospects and job plans. On the whole, males' satisfaction with their job status is higher than that of females.

The analysis found that the advantage of females in the publishing industry did not provide them with better pay, benefits or career development opportunities than males. Gender equality is only manifested in workload, but in other aspects is just an empty slogan. Even when better educated and younger than males, females are still at a disadvantage when it comes to getting promotions and pay rises. This also reduces females' expectations for future career development, with females having an overall lower career satisfaction level than males. In short, in the current publishing industry, there is an obvious gender gap, and males still have an absolute advantage.

## 5. Discussion

Through the research on the working status and career development of Chinese publishing practitioners, this paper finds that in the current social context of China, gender discrimination has hindered the normal upward flow of females in the professional field to a certain extent, and has reduced females' career expectations and satisfaction. There is also a strong "glass ceiling" effect on females' career development. In combination with relevant theories and the basic situation of China's society, we will analyze the causes of the gender gap in the publishing industry and the glass ceiling effect from the following three points.

### 5.1. The Influence of Stereotype

Gender stereotypes describe what males and females should do and how they should act. It is a cognitive shortcut that will affect people's processing of information related to males and females, including description and convention. Descriptive gender stereotypes refer to what males and females are. They believe that the mismatch between females'

appearance and their attributes leads to difficulty in gaining success in male gender-type positions and roles. Conventional gender stereotypes mainly describe what males and females should be. Prescriptive gender stereotypes promote gender bias by setting normative standards for behavior. If the standards are violated, social punishment will be incurred. Among them, the stereotypical effect of gender stereotyping believes that females' success over males is a violation of convention and should be punished. These stereotypes become a special obstacle for females' career promotion [22]. Gender stereotypes have greatly restricted the social status and self-positioning role cognition of intellectual females in China. In the workplace, there is a bias of male dominance and female subordination, and even females themselves admit such a role determination. Many males disdain accepting the leadership of high-level females or are unwilling to accept the feminized leadership style. High-level positions are generally dominated by men. Some male personnel in higher vocational roles are unwilling to involve females in higher vocational education and tend to suppress females. Due to there being a majority of male leaders in middle and senior management positions in most enterprises, "homogeneous social reproduction" leads males to prefer to promote male employees who are more similar to themselves, rather than leave opportunities to female employees who are more obviously different to themselves. As a result, the tradition of males serving as corporate executives has continued [23], and females' career development lacks its due opportunities.

### 5.2. The Impact of Gender Division and Traditional Concepts

The gender division of labor based on the physiological characteristics of males and females has been internalized into a gender culture over time. The survey on the social status of Chinese females (2011) showed that 61.6% of males and 54.8% of females agreed with the view that "men should focus on society and females should focus on family" [24]. The family norms that have formed over a long time believe that males should work hard for their careers, while females should pay more for their families, which is the result of gender division. Females not only bear more family responsibilities than men, but also want to be good wives and mothers at the same time. Therefore, they face the dual pressure of career and family. Many professional females have experienced the phenomenon of the "female middle-aged crisis". When career development and family care conflict, females are more often expected to give up work for their families. Once they encounter the "glass ceiling", they easily give up fighting and choose work that is easier in order to take their families into account [25]. Some people, therefore, stop at the "glass ceiling" and stay at the middle level. If females' long-term development in the workplace is not smooth, they will gradually believe that "doing well is better than marrying well" in society. The number of females who agree with this view increased from 37.3% in 2000 to 48.0% in 2010 [24,26]. Crosby et al. The influence of gender culture and long-term traditional culture has led females to face unfavorable "cultural support" and significantly lower expectations than males in their career development, which is also one of the main reasons for the difficulty of females' promotion.

### 5.3. Policy Adjustment Imbalance

First of all, from the perspective of public employment policy, China's retirement age is 60 for males and 55 for females [27]. This policy is undoubtedly harmful to females' career development and economic interests. Moreover, for maternity leave and parental leave, the legal maternity leave for females is 128 days, which can be extended to 218 days according to specific conditions, while paternity leave for males is 15 days [28]. This regulation seems to be aimed at taking care of females because females need a period to recover after giving birth. However, in the workplace, it also puts females at a disadvantage in professional competition. In addition, some employment policies that favor females often fail to protect females' career development due to misinterpretation in the formulation and implementation; however, these exacerbate the isolation of the glass ceiling. This study confirms that there is no significant difference in the workload between males and

females in the publishing industry. In fact, in modern society, which is characterized by the knowledge economy, the difference in the physical fitness of males and females is no longer the biggest obstacle to females' career advancement. Secondly, in terms of law, although China has attached great importance to the issue of equal employment between males and females since the founding of the People's Republic of China, a large number of laws and policies have been issued to promote the development of females' employment, so that females have the right to employment and gain economic independence. Under the conditions of market economy, the government has also formulated many policies to protect females' rights and interests, and promote the development of females' employment. However, it is undeniable that the current mechanism has temporary characteristics and lacks an effective punishment mechanism, which greatly reduces the effectiveness of the implementation of legal policies.

To sum up, due to the comprehensive effect of various factors, as a female-led cultural service industry, the publishing industry has not fully reflected the equal rights of males and females. It is of great significance and value to explore and seek gender equality. Governments, industries, associations and publishing institutions should also pay more attention to providing female publishers with equal career choices and career development paths. Therefore, females can get the same results as males after working hard.

Admittedly, this paper has improved our understanding of the current situation of the gender gap among publishing practitioners, but there are still some deficiencies in this study. Because the data used in the study are cross-sectional data, the impact of other factors on employment status and career development cannot be completely excluded. The current research is still unable to understand the mechanism behind various factors, and some analysis results do not reflect the impact of potential mediators. These issues need to be further explored in future research, in order to increase our understanding of the social factors that contribute to gender differences in career promotion.

**Author Contributions:** Y.L.; methodology, software, formal analysis, writing. Y.Z.; data collection, supervision, modification. All authors have read and agreed to the published version of the manuscript.

**Funding:** This research received no external funding.

**Data Availability Statement:** The following are available online at: https://github.com/Ivy-yawen/gender_gap (accessed on 1 December 2022) for questionnaire and data for the questionnaire, data and original results.

**Acknowledgments:** Thanks to Nairui Xu for her suggestions on the structure and grammar of this article.

**Conflicts of Interest:** The authors declare no conflict of interest.

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
