# Peer review of "The Gender Gap in Job Status and Career Development of Chinese Publishing Practitioners"

_publications, doi:10.3390/publications11010013_

Round 1

Reviewer 1 Report

This paper is well written. Clarity of the explanation cannot be denied

But this paper is mainly descriptive. Main weakness :

-lack of theoretical framework ( vertical segregation theories / glass ceiling / concept of career / Gender discrimination) could have been used in this paper) 

-lack of references in the discussion, especially when focusing on China"s traditional concepts, policies, regulations and gender bias.  How the collected data confirm or not the current litterature on gender discrimination / glass ceiling effect in the publishing sector ?

confront your findings with the main theoretical framework.

suggestion : i would use Jobs perception instead of job emotions

Reviewer 2 Report

The paper entitled The gender gap in job status and career development of Chinese publishing practitioners addresses an interesting and hot topic related to gender gap analysis in different fields of activity. It is generally well structured and written, although some improvement should be done:

- the introduction part is poorly documented, especially Related work section. Gender gap issue is a topic well debated in the literature and different points of view should be presented in the text. More references should be studied and added, including debates on other fields of activity: 

Luisa S Flor et al. Quantifying the effects of the COVID-19 pandemic on gender equality on health, social, and economic indicators: a comprehensive review of data from March, 2020, to September, 2021, The Lancet, 2022, https://doi.org/10.1016/S0140-6736(22)00008-3.

Stefan, D. et al. 2021. Women Entrepreneurship and Sustainable Business Development: Key findings from a SWOT-AHP Analysis. Sustainability. https://doi.org/10.3390/su13095298

Global Gender Gap Report 2022, World Economic Forum

Adachi, T.; Hisada, T. Gender differences in entrepreneurship and intrapreneurship: An empirical analysis. Small Bus. Econ. 2017

- Materials and Methods section should present statistics related to the number of possible respondents to check the relevance of the sample. Also, the questionnaire should be briefly described: types of questions, structure, etc.  

Round 2

Reviewer 1 Report

The paper has been greatly improved

Thank you to the authors